# Identification and Quantification of Selected Benzoxazinoids and Phenolics in Germinated Spelt (*Triticum spelta*)

**DOI:** 10.3390/foods12091769

**Published:** 2023-04-24

**Authors:** Andrej Živković, Dejan Gođevac, Blaž Cigić, Tomaž Polak, Tomaž Požrl

**Affiliations:** 1Department of Food Science and Technology, Biotechnical Faculty, University of Ljubljana, SI-1111 Ljubljana, Slovenia; andrej.zivkovic@bf.uni-lj.si (A.Ž.); blaz.cigic@bf.uni-lj.si (B.C.); tomaz.polak@bf.uni-lj.si (T.P.); 2Institute of Chemistry, Technology and Metallurgy, National Institute of the Republic of Serbia, University of Belgrade, 11000 Belgrade, Serbia; dejan.godjevac@ihtm.bg.ac.rs

**Keywords:** spelt, germination, benzoxazinoids, *cis*-isomers, schaftoside, free and bound fractions

## Abstract

In this study, we investigated the effects of germination on the secondary metabolite composition in spelt grains. Germination significantly increased the content of various metabolites in free and bound forms. Benzoxazinoids were the most important compounds in the free fraction of the 96 h germinated grains (MBOA content as the predominant compound was 277.61 ± 15.29 µg/g DW). The majority of phenolic acids were present in the bound fraction, with *trans*-ferulic acid as the main component, reaching 753.27 ± 95.87 µg/g DW. The often neglected *cis*-isomers of phenolic acids accounted for about 20% of the total phenolic acids. High levels of apigenin di-C-glycosides were found in spelt grains, and the schaftoside content was most affected by germination, increasing threefold. The accumulation of secondary metabolites significantly increased the antioxidant activity of germinated spelt. According to the results of this study, the content of most bioactive compounds was highest in spelt grains after 96 h of germination. These data suggest that germinated spelt could potentially be valuable for the production of functional foods.

## 1. Introduction

Spelt (*Triticum spelta* L.) is primitive wheat that is a distant cousin of common wheat (*Triticum aestivum* L.). Along with einkorn (*Triticum monococcum* L.) and emmer (*Triticum turgidum* L.), spelt is considered an ancient wheat that has remained unchanged over a very long period of time. Compared to common wheat, ancient wheats are more resistant to disease, require less nitrogen fertilization, and are generally better adapted to harsh growing conditions. Products from spelt and other ancient cereals are reported to be better tolerated by individuals with intolerances or allergies to modern wheat varieties [1]. Recent evidence from in vitro, in vivo, and clinical studies suggests that the consumption of ancient wheat products has antioxidant and anti-inflammatory effects. In addition, ancient cereals and their products have a lower glycemic index than other grains [2].

As a result of these findings and its image as a “healthier and more natural” cereal compared to modern wheat varieties, spelt is gaining popularity in both conventional and organic agriculture and thus commercial interest in the food industry.

Germination processes are a traditional method for improving the nutrient profile of grains and offer a practical way to naturally biotransform grains. It is considered a “green food” engineering method to accumulate natural bioactive compounds in seeds and sprouts that can be consumed as functional foods [3]. Germination is initiated by increasing the moisture content of the grain to 43–45% by soaking in water [4]. After initiation, storage macromolecules are degraded by newly synthesized enzymes, and these reactions lead to the development of new highly bioactive compounds through de novo synthesis and transformation, thus increasing the nutritional value and health-promoting effects of germinated grains [5]. Germination increases various metabolites (polyphenols, alkylresorcinols, vitamins) and many other less known groups of secondary metabolites. Other benefits of germination include removal or reduction of antinutritional compounds (e.g., phytates) and enrichment with dietary fiber [3,6,7].

In the literature, phenolic compounds are the most reported type of bioactive compounds in cereals, and they generally occur in free and bound forms. However, the total phytochemical content in whole grains is often underestimated since most studies determine only the content of free phenolics. The free form of phenolic compounds accounts for only a small portion of the phenolics in grains [8]. The majority of phenolics in cereals are bound to cell wall materials, such as lignin, cellulose, proteins, and arabinoxylan and can be released only by alkaline or enzymatic hydrolysis [9]. Consequently, bound phenolic compounds can survive gastrointestinal digestion to reach the colon intact, where they may provide a favorable antioxidant environment for intestinal microbiota [10]. This might partly explain the positive health effect of whole-grain consumption, as demonstrated by epidemiological studies [11]. In addition to phenolic acids, there are other, often neglected compounds found in cereal grains, such as benzoxazinoids and apigenin di-C-glucosides (mainly schaftoside and isoschaftoside). It is suggested that these compounds could be additional contributing factors to the health benefits of whole grains.

Benzoxazinoids are nitrogen-containing secondary metabolites found mainly in the vegetative parts of Poaceae plants, such as rye, wheat, triticale, and maize. They are usually divided into three groups according to their structure: benzoxazolinones, lactams, and hydroxamic acids. Benzoxazinoids are mostly analyzed because of their importance in plant physiology as allelochemicals for defense against predators and infection [12]; however, to the best of our knowledge, there is no information about dynamic changes in benzoxazinoids during germination of spelt grains or in spelt grains whatsoever.

Recent studies have found that benzoxazinoids are present not only in the vegetative parts of young plants but also in mature cereal grains. Considerable amounts of benzoxazinoids are found in fermented beverages made from wheat or rye malt [13]. With the significant dietary intake of various whole grain cereals and cereal products, the potential health-promoting effects of benzoxazinoids have come into focus.

There are still no conclusive data on the health effects of these compounds in humans [14]; however, recent studies on benzoxazinoids have reported their pharmacological and health-promoting properties, including anti-inflammatory, anticancer, antimicrobial, appetite suppressant, and reproductive system stimulant effects [15]. Animal and human studies have demonstrated that dietary benzoxazinoids are absorbed and metabolized in mammals [16]. Quantitative analysis of benzoxazinoid metabolism in a rat model revealed that the bioavailability of various benzoxazinoids based on their urinary excretion levels varied from <1 to 21% [17]. The pharmacokinetics of benzoxazinoids in plasma and urine have shown that benzoxazinoid levels are dose-dependent, and it takes approximately 3 h to reach the highest plasma levels. After consumption, most benzoxazinoids and their metabolic derivatives are excreted slowly in the urine [18].

Flavonoids are another group of compounds present in cereal seeds and represent a large family of plant polyphenolic compounds that act as UV filters and colorants, serve to defend against pathogens, and are of great interest to human health. Schaftoside and isoschaftoside are flavonoid di-C-glycosides that possess a variety of biological activities, including antidiabetic, antihypertensive, hepatoprotective, anti-inflammatory, and antioxidant activities in mammals, with potential for applications as drugs or dietary supplements [19,20]. Because of the various health benefits of compounds in cereal grains, understanding the dynamic changes in bioactive compounds in cereals, both in free and bound forms, during germination is highly important [21]. However, limited information is available on the dynamic changes in benzoxazinoids and apigenin di-C-glycosides during the first stages of germination.

The present investigation reveals the effect of germination on the quantitative contribution of these substances to the total nutritional value of spelt grains. Therefore, the main objective of this study was to study and characterize the dynamic changes in the composition and content of selected phenolics, benzoxazinoids, and apigenin di-C-glycosides in free and bound forms of spelt grains at different stages of germination. The second objective was to quantify often neglected cis isomers of phenolic acids in the bound fraction of spelt extracts. Finally, we evaluated the effect of germination on the total phenolic content and antioxidant activity. Our goal was to provide information that can contribute to the implementation of germinated spelt grains for the production of new functional food products.

## 2. Materials and Methods

### 2.1. Materials

Throughout this work, we used the »Ostro« cultivar of common spelt (*Triticum spelta* L.) grown in organic growth conditions that were kindly supplied by Rangus mill (Šentjernej, Slovenia). Methanol, hydrochloric acid, sodium bicarbonate, sodium hydroxide, 1,1-diphenyl-2-picrylhydrazyl (DPPH), and 2,2’-azino-bis (3-ethylbenzothiazolin-6-sulfonic acid diammonium salt) (ABTS) were purchased from Sigma-Aldrich (Steinheim, Germany). Manganese dioxide was from Kemika (Zagreb, Croatia) and Folin–Ciocalteu (FC) reagent was from Merck (Darmstadt, Germany). The analytical standards of 6-methoxy-2-benzoxazolinone (MBOA), 2-benzoxazolinone (BOA), ferulic acid, p-coumaric acid, and a mixture of *cis*/*trans* isomers of ferulic acid were from Sigma-Aldrich (Steinheim, Germany). Schaftoside and isoschaftoside were from Biosynth (Bratislava, Slovakia). All of the standards used were analytical or HPLC grade. All aqueous solutions were prepared using Milli-Q purified water (Merck Millipore, Bedford, MA, USA).

### 2.2. Germination Method

Before germination, we prepared spelt samples by removing foreign materials and damaged grains. Disinfection of the grains was performed by soaking the samples in 50 °C water for 5 min [22]. The soaking time was 8 h in water, with a 15 min aeration period every hour. After soaking, the grains were placed in a thin layer on perforated metal trays. The trays were placed in a growth chamber with a humidifier to ensure high humidity (relative humidity, >95%) and germinated at 20 °C. Germinated seeds were harvested at 12, 24, 36, 48, 72, and 96 h after the start of soaking. Prior to analysis, moisture content in all samples was determined using a modified AACC method 44-19.01, and results were expressed on a dry weight basis.

### 2.3. Extraction of Free Phenolic Compounds

The extraction of free phenolic compounds was carried out according to the method previously reported by Živković et al. [7]. Briefly, 1 g of ground grains was mixed with 3.0 mL 70% aqueous methanol, and the mixture was shaken in the dark at room temperature for 40 min at 200 rpm (EV-403; Tehtnica Železniki, Železniki, Slovenia). After centrifugation at 8709× *g* for 8 min at 10 °C (Avanti JXN-26; Beckman Coulter, Krefeld, Germany), the supernatant was removed and stored, and the extraction was repeated twice more. The 3 supernatants were pooled, diluted to 10 mL with 70% aqueous methanol, filtered using 0.45-µm pore size syringe filters (Chromafil A-45/25; Macherey-Nagel, Düren, Germany), and stored at 2 °C until determination of total phenolic content (TPC) and antioxidative activity (AA) analysis within 24 h.

### 2.4. Extraction of Bound Phenolic Compounds

After methanol extraction, the solid residues were hydrolyzed with sodium hydroxide, as described previously by Živković et al. [7]. Briefly, 20 mL of 2 M NaOH was added to the reaction tube, and the mixture was shaken in the dark at room temperature for 4 h at 200 rpm (Tehtnica Železniki EV-403, Slovenia). The hydrolyzed mixture was acidified to pH 3.2 to 3.4 by the addition of 3.5 mL of concentrated formic acid. After centrifugation at 8709× *g* for 8 min at 10 °C (Avanti JXN-26; Beckman Coulter, Krefeld, Germany), the supernatant was removed and filtered through 0.45-µm pore size syringe filters (Chromafil A-45/25; Macherey-Nagel, Düren, Germany) and stored at 2 °C until determination of total phenolic content (TPC) and antioxidative activity (AA) analysis within 24 h.

### 2.5. Total Phenolic Content (TPC)

The TPC of the crude grain extracts was determined using the Folin–Ciocalteu (FC) reagent according to Živković et al. [7]. A total of 100 microliters of extract was dispensed into 2.0 mL microcentrifuge tubes and mixed with 1.3 mL Milli-Q water and 0.3 mL diluted FC reagent (reagent:water, 1:2). After 5 min, 0.3 mL 20% (*w*/*v*) aqueous Na_2_CO_3_ was added. After 1 h at room temperature, the absorbances were measured at 765 nm (UV-Vis spectrophotometer; Model 8453; Agilent Technologies, Santa Clara, CA, USA). The measurements were compared with a standard curve of a Trolox solution, and TPC was expressed as mg Trolox equivalents (TE) per g dry weight of the grain sample (mg TE/g DW). Trolox was used as a standard because it exchanges the same number of electrons in FC, ABTS, and DPPH assays [23], which allows a direct comparison of the relative efficiency of extracted phenolic compounds in FC and antioxidant assays.

### 2.6. DPPH Radical Scavenging Activity

The DPPH radical scavenging activity [24] of the grain extracts was determined according to a method described previously [7]. A total of 50 microliters of each extract was mixed with 250 µL of acetic buffer, 0.7 mL of 70% aqueous methanol, and 1 mL of a 0.2 mM methanol solution of DPPH to give a final volume of 2 mL. The absorbance of the mixture was measured after 1 h at 517 nm (UV-Vis spectrophotometer; Model 8453; Agilent Technologies, Santa Clara, CA, USA). The measurement was compared to a standard curve of a Trolox solution, and the radical scavenging activity was expressed as mg Trolox equivalents per g dry matter (mg TE/g DW).

### 2.7. ABTS Radical Cation Scavenging Activity

The ABTS radical scavenging activities of the grain extracts were determined according to a method described previously [7]. A total of 50 microliters of each extract was mixed with 0.5 mL 0.325 M phosphate buffer, 1.0 mL of diluted ABTS radical cation solution, and 0.45 mL Milli-Q water to give a final volume of 2 mL. The mixture was shaken and left in the dark for 1 h. The absorbance was measured after 1 h at 734 nm (UV-Vis spectrophotometer; Model 8453; Agilent Technologies, CA, USA). The measurement was compared to a standard curve of a Trolox solution, and the radical scavenging activity was expressed as mg TE/g DW.

### 2.8. Purification of Extracts

Crude grain extracts were purified using 100 mg Strata-X RP cartridges (Phenomenex, Torrance, CA, USA) according to a previously described method [7]. Briefly, 30 mL of the diluted crude methanol extracts (extract:water, 1:9) or 3.0 mL of the hydrolyzed extracts were applied to the SPE cartridges, washed with 4.0 mL of Milli-Q water, and dried with a flow of air. The compounds bound to the cartridges were eluted with 2.0 mL of 70% (*v*/*v*) aqueous methanol. The resulting extracts were filtered through syringe filters with a pore size of 0.20 µm (Chromafil Xtra-20/13; cellulose acetate; Macherey-Nagel, Düren, Germany) and then stored at −80 °C until liquid chromatography–mass spectrometry analysis (LC-MS).

### 2.9. Liquid Chromatography–Mass Spectrometry Analysis

For separation and quantification of each compound in spelt extracts, reversed-phase LC-MS analysis was used. The LC system used (1100 chromatography system; Agilent Technologies, CA, USA) included a thermostated autosampler (G1330B), a thermostated column compartment (G1316A), a diode array detector (G1315B), and a binary pump (1312A). The LC system was coupled with a mass spectrometer (Quattro micro API; Waters, Milford, MA, USA). Chromatographic separation was carried out using a C18 column (2.7 μm, 150 mm × 2.1 mm; Ascentis Express) with a C18 guard column (2.7 μm, 5 mm × 2.1 mm; Ascentis Express; Supelco, Bellefonte, PA, USA). The conditions used were as follows: column temperature, 35 °C; injection volume, 2 µL; and mobile phase flow rate, 320 µL/min. The components of the mobile phase were 0.1% aqueous formic acid (solution A) and acetonitrile (solution B). The mobile phase gradient was programmed as follows (%B): 0–4 min, 10%; 4–18 min, 10–60%; 18–18.2 min, 60–80%; 18.2–20 min, 80%; 20–20.2 min, 80–10%; and 20.2–26 min, 10%. Detection was performed with scanning diode array spectra from 240 nm to 650 nm.

The mass spectrometer was operated in negative ionization mode, and the operating conditions were as follows: electrospray capillary voltage, 3.5 kV; cone voltage, 20 V; extractor voltage, 2 V; source block temperature, 100 °C; desolvation temperature, 350 °C; cone gas flow rate, 30 L/h, and desolvation gas flow rate, 350 L/h. The data signals were acquired and processed on a PC using MassLynx software (V4.1 2005; Waters Corporation, Milford, MA, USA). Compared to previously determined calibration curves, identification of the individual compounds was achieved by comparing their retention times and both the spectroscopic and mass spectrometric data, with quantification according to peak areas.

The compound corresponding to peak 16 in the LC-MS chromatograms was isolated by repetitive semipreparative chromatography runs. The chromatography conditions and gradient were the same as specified previously in the main experiment.

### 2.10. NMR Spectroscopy

NMR spectra were recorded on a Bruker Avance III 500 NMR spectrometer with a proton NMR frequency of 500.26 MHz and a 5 mm BBO probe head using standard pulse sequences. Methanol-d4 was used as the solvent, and spectra were recorded at 298 K. The spectra were referenced to the residual proton signal of methanol-d4. Chemical shifts were given in δ (ppm), and coupling constants were given in Hz.

### 2.11. Statistical Analysis

All experiments were carried out in triplicate using a complete randomization method. All spelt extracts were prepared in duplicate. Data were presented as mean ± standard deviation (SD) for three analyses for each extract. Results were subjected to two-way comparison ANOVA, and significance of differences between means was determined using Tukey’s Multiple Comparison Tests. Data analysis was performed using SPSS Statistics software (version 24; IBM, New York, NY, USA). Statistical significance was defined at the level of *p* < 0.05.

## 3. Results and Discussion

### 3.1. Effect of Germination on Total Phenolic Content and Antioxidant Activity

The changes in the content of free and bound phenolic compounds at the different germination stages are shown in Table 1. Our results show that germination induced dynamic changes in free, bound, and total phenolic compounds in spelt. Germination had a significant effect on the content of free phenolic compounds (*p* < 0.05), which increased from 1.17 mg TE/g DW in nongerminated grains to 4.57 mg TE/g DW at the end of the 96 h germination period. Bound phenolic compounds were the predominant form in the nongerminated grains and accounted for approximately 72.0% of the TPC on a dry weight basis. This is consistent with results previously reported for TPC in nongerminated spelt [25,26]. It was found that germination increases the TPC of both free and bound phenolic compounds in spelt grains, so the next objective was to evaluate the dynamic changes in antioxidant capacity of the free and bound fractions of germinated spelt. The spelt extracts were evaluated for their scavenging activity against the stable free radicals DPPH and ABTS. The radical scavenging activity of DPPH and ABTS radicals in germinated spelt extracts was expressed as milligrams of Trolox equivalents per gram dry weight (mg TE/g DW). The results are summarized in Table 1. With both assays, the lowest AA was measured in the free fraction of grains prior to germination (Table 1). Similar values of AA in nongerminated spelt were reported by Yilmaz et al. [26]. In the same study, the levels of AA measured by the ABTS assay were 6–8-fold higher than the levels of AA measured by the DPPH method. As seen from our results in Table 1, similar differences were found between the values measured by the DPPH method and the ABTS values. The differences in the measured antioxidant capacity between the two assays can be explained by the different reactivities (Appendix A) of the compounds present in the spelt extract toward the free radicals ABTS and DPPH [27].

The antioxidant activity in both free and bound fractions gradually increased during the observed 96 h germination period. These results are in agreement with previous reports in other cereals, such as rice [28], wheat [29], and buckwheat [7]. As seen from the results of TPC and AA of the germinated spelt extracts, the ratio between the measured values in the free and bound fractions changed during the germination process (Figure 1). This difference is most pronounced for values obtained by the FC method. Free phenolic compounds in nongerminated spelt extracts accounted for 28% of the total phenolic content, while this ratio increased to 50% in extracts of spelt germinated for 96 h. The change in the ratio between free and bound phenolics was also measured by the DPPH and ABTS assays but not as much as by the FC assay. This may be explained by the accumulation of newly synthesized compounds in the germinated spelt and their different reactivities in the FC, DPPH, and ABTS assays. According to the profile of compounds from LC-MS analysis, the major constituents in the free fractions of germinated spelt were benzoxazinoids. The specific reactivities of BOA and MBOA toward the FC, DPPH, and ABTS reagents (Appendix A) were analyzed, and the results showed that MBOA had a high affinity for the FC reagent and almost no affinity for either ABTS or DPPH. Differences in the reactivity of MBOA and probably other benzoxazinoids to FC, DPPH, and ABTS assays can explain the difference in the ratio between the measured values of free and bound fractions in nongerminated and germinated spelt extracts.

### 3.2. Phenolic Characterization by Liquid Chromatography–Mass Spectrometry

LC-MS analysis of the extracts of the raw and germinated grains (i.e., sprouts) revealed the presence of several compounds in free and bound forms. The changes in the profile of the compounds of germinated spelt at different stages of germination are shown in Table 2, and the representative chromatograms of the detected components are shown in Figure 2. The results show large differences between the compounds in germinated and nongerminated spelt.

To identify some of the detected compounds, we used chromatographic separation to isolate a selected peak (peak 16) and NMR spectroscopy for structural identification. The isolated peak was identified by NMR spectroscopy as 6-methoxy-2-benzoxazolinone (MBOA) (Appendix A). Based on the identified compound, another benzoxazinoid, 2-benzoxazolinone (BOA, peak8), was identified by comparing UV-Vis spectra (Appendix A), MS data, and retention times with the commercial standard. Since standards for other benzoxazinoids were not commercially available, Peak 5 was characterized based on UV-Vis spectra (Appendix A) and MS data and tentatively identified as 2-hydroxy-1,4-benzoxazin-3-one (HBOA) and was expressed as MBOA equivalents (µg/g extract). We also detected several unidentified compounds in the free fraction. Peaks 2 (*m*/*z* 388.2), 4 (418.2), 6 (*m*/*z* 594.5), and 14 (*m*/*z* 432.4) had UV-Vis spectrum profiles similar to that of HBOA (Appendix A) and similar to UV-Vis spectra of other lactams and hydroxamic acids in the literature (λmax = 264–266 nm) [30,31,32]. We can assume that these compounds may be related to benzoxazinoids, more specifically to lactams, hydroxamic acids, or their methyl derivatives, but due to a lack of literature data and no available commercial standards, we cannot confirm this hypothesis. MBOA was detected 24 h after soaking, and during 48 to 96 h germination, its content gradually increased to maximum values of 277 µg/g DW. BOA also increased during germination, but the concentrations were much lower than those of MBOA. The content of HBOA was strongly affected by germination and reached 219.65 µg/g DW at the end of the germination period. This strong increase in benzoxazinoids in the early stages of germination is consistent with previous studies of metabolic synthesis of benzoxazinoids in rye [33] in which genes responsible for benzoxazinoid synthesis showed the highest expression levels 24–30 h after the onset of germination. Although there are no available data yet on the biosynthesis of benzoxazinoids in spelt, we can assume that pathways similar to those in other Poaceae plants are initiated during germination.

Two phenolic acids were detected in methanol extracts but at very low concentrations. In nongerminated grains, ferulic acid was the only phenolic compound detected (1.9 and 0.96 µg/g DW for *trans* and *cis* isomers, respectively). In germinated grains, *trans*-p-coumaric acid was also detected, with the highest concentration reached at the end of germination (1.59 µg/g DW).

Characterization of the methanol extracts in the grains before germination showed a significant content of a compound with an *m*/*z* of 563.5 (peaks 9, 11, and 12). Based on retention times and fragmentation pattern matching with a commercial standard (Appendix A), the compounds were identified as apigenin-di-C-glucosides (schaftoside and its structural isomers). Literature data concerning schaftoside and its isomers in wheat and cereals are relatively scarce [34], and to the best of our knowledge, there is no information about dynamic changes in apigenin-di-C-glucosides during germination. Of the three apigenin di-C glucosides detected in nongerminated spelt, schaftoside was detected at the lowest concentration (13.58 µg/g DW), but it was most affected by germination, reaching 53.69 µg/g DW at the end of the 96 h germination period. Although the content of most compounds in spelt increased during germination, the main compounds in raw grains (structural isomers of schaftoside) were not significantly affected by the germination process. Flavonoids in plant tissues are associated with protection against UV light, and their content usually increases when the plant is exposed to UV radiation [35]. Since the germination process in our experiment was conducted under dark conditions, it is possible that there was no activation of the photoreceptors responsible for the induction of UV-protective flavonoid synthesis [36].

A higher concentration of phenolic compounds was detected in the bound fraction of spelt grain extracts. As expected from the literature data [37], the most abundant phenolic compounds in the bound fraction were *trans*-ferulic acid and *trans*-p-coumaric acid. Significant accumulation of bound phenolic compounds during germination was also observed. This confirms the conclusions reached in the literature [38,39] that during germination, the content of bound phenolics increased gradually. This increase coincides with the growth of the seedling and the development of new tissue. Previous studies reported high levels of bound phenolics in leaves and other tissues of cereal plants [40], so this increase can be explained by the growth of tissues (shoot and radicle) rich in bound phenolics. The content of *trans*-ferulic acid increased significantly and reached 753 µg/g DW after 96 h of germination, which corresponds to an increase of 110% compared to raw grains. *Trans*-p-coumaric acid, which was also present in high concentrations in the bound fraction, showed a significant increase during germination (*p* < 0.05), reaching 119.77 µg/g DW (610% increase) after 96 h of germination. In addition to *trans*-ferulic acid and *trans*-p-coumaric acid, two other peaks were detected in the bound fraction (peaks 10 and 15). The MS analysis of the samples showed that Peak 10, eluted 1 min after *trans*-p-coumaric acid, had the same *m*/*z* ratio as *trans*-p-coumaric acid (*m*/*z* 163). The unknown compound showed the same fragmentation pattern as *trans*-p-coumaric acid, so we can assume that the peak occurring after *trans*-p-coumaric acid is the corresponding *cis*-isomer. Peak 15 was confirmed to be *cis*-ferulic acid, according to the retention time of the commercial standard of the ferulic acid isomer mixture and MS data analysis. A similar conclusion was reached by another study [41], where secondary peaks of p-coumaric, caffeic, and ferulic acid were noticed. The two *cis*-isomers were quantified using the calibration curve of their respective *trans*-isomers. The levels of *cis*-isomers were also significantly increased after germination and were 165% and 477% higher than the levels of *cis*-ferulic acid and *cis*-p-coumaric acid in nongerminated grains, respectively. Our results provide a new perspective on phenolic acid content in cereals. Most of the works on bound phenolic content in cereals did not mention the existence of *cis*-stereoisomers of phenolic acids due to the lack of effective separation methods. Only a few papers mention the separation and identification of *cis*-stereoisomers in food samples [42,43]. In the present study, *cis*-ferulic acid accounted for approximately 20% of the total ferulic acid content in the spelt samples, which is consistent with the results reported by Tang et al. [42]. The proportion of *cis*-p-coumaric acid was lower and accounted for approximately 8% of the total p-coumaric acid content. This proportion of *cis*-stereoisomers of phenolic acids in cereals opens new questions about the possible physiological functions of *cis*-stereoisomers in biological systems.

According to the data in the present study, germination of the spelt grains had significant effects on the profile of secondary metabolites, as well as the levels of individual compounds in germinated grains. These results provide important information on the nutraceutical quality of germinated spelt grains.

## 4. Conclusions

The germination process resulted in important changes in the composition of spelt grains. The total phenolic content, antioxidant activity, and secondary metabolite content were significantly increased during germination. Ferulic and p-coumaric acids (*cis* and *trans* forms combined) were the major phenolic acids in spelt grains, and their total content increased from 457.93 ± 12.03 µg/g DW in nongerminated spelt to 1090.99 ± 101.99 µg/g DW in germinated spelt. Benzoxazinoids and apigenin-di-C-glucosides are often neglected compounds in cereals, and germination strongly affects their content in spelt grains, as the sum of the three quantified benzoxazinoids reached a concentration of 517.75 ± 48.49 µg/g DW in germinated spelt, while no benzoxazinoids were detected in nongerminated spelt. Among apigenin-di-C-glucosides, only schaftoside was significantly affected (nearly four-fold increase during germination). According to the results of the present study, the content of most bioactive compounds was highest in spelt grains after 96 h of germination. Obtained data are promising for the high-value application of germinated spelt in functional foods (e.g., enriched baked goods, whole seed flowers, ready-to-eat snacks, etc.), but further studies on the health benefits of spelt secondary metabolites are essential to better understand their health-promoting effects on humans. Overall, germinated edible grains and sprouts rich in bioactive compounds can be considered as an important raw material for the production of functional foods that have a positive impact on the prevention of some chronic diseases.

## Figures and Tables

**Figure 1 foods-12-01769-f001:**
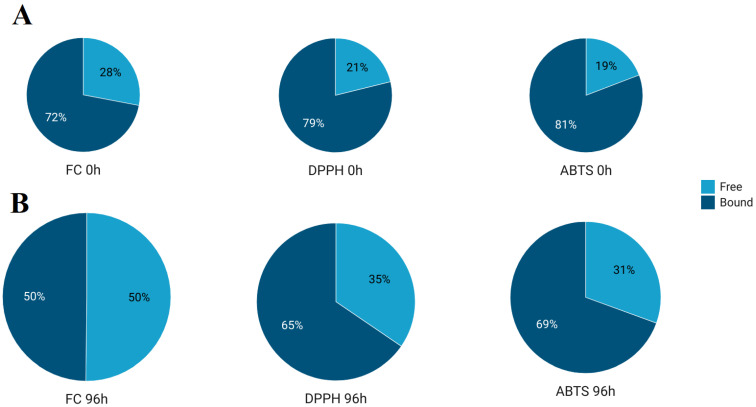
Ratio between values of free and bound fractions measured by FC, DPPH, and ABTS assays in nongerminated spelt (**A**) and 96 h germinated spelt (**B**). An increase in plot area represents a relative increase of measured values during 96 h of germination.

**Figure 2 foods-12-01769-f002:**
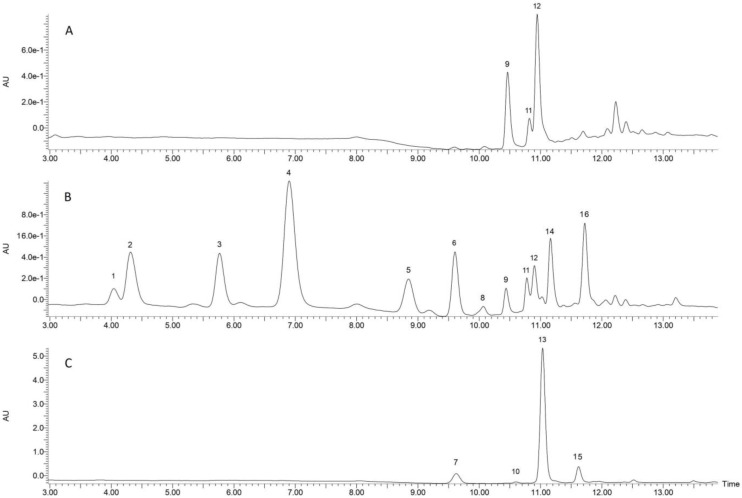
Chromatograms obtained through LC-MS analysis of (**A**) free fraction of nongerminated spelt, (**B**) free fraction of 96 h germinated spelt, and (**C**) bound fraction of 96 h germinated spelt. Peaks: 1–unidentified, (*m*/*z* 534.3); 2–unidentified, (*m*/*z* 388.2); 3–unidentified, (*m*/*z* 134.2); 4–unidentified, (*m*/*z* 418.2); 5–HBOA; 6–unidentified, (*m*/*z* 594.5); 7–*trans* p-coumaric acid; 8-BOA; 9–schaftoside structural isomer 1; 10–cis p-coumaric acid; 11–schaftoside; 12–schaftoside structural isomer 2; 13–*trans* ferulic acid; 14–unidentified, (*m*/*z* 432.4); 15–cis ferulic acid; 16–MBOA.

**Table 1 foods-12-01769-t001:** Total phenolic content (FC) and antioxidant activities (DPPH, ABTS) for the free and bound extraction fractions of the spelt extracts during germination.

Analysis	Measure	Germination Time (h)						
		Nongerminated	12	24	36	48	72	96
Total phenolic content (mg TE/g dry weight)								
FC	Free	1.17 ± 0.08 ^a^	1.39 ± 0.06 ^ab^	1.47 ± 0.07 ^ab^	1.76 ± 0.11 ^b^	2.40 ± 0.15 ^c^	3.12 ± 0.28 ^d^	4.57 ± 0.57 ^e^
	Bound	3.01 ± 0.16 ^a^	3.12 ± 0.24 ^ab^	3.35 ± 0.28 ^ab^	3.45 ± 0.57 ^ab^	3.69 ± 0.29 ^b^	4.34 ± 0.20 ^c^	4.54 ± 0.40 ^c^
	Total	4.18 ± 0.18 ^a^	4.51 ± 0.29 ^a^	4.82 ± 0.30 ^a^	5.21 ± 0.59 ^ab^	6.09 ± 0.41 ^b^	7.46 ± 0.46 ^c^	9.11 ± 0.96 ^d^
Antioxidant activity (mgTE/g dry weight)								
DPPH	Free	0.36 ± 0.03 ^a^	0.43 ± 0.05 ^ab^	0.50 ± 0.05 ^b^	0.61 ± 0.04 ^c^	0.72 ± 0.06 ^d^	0.92 ± 0.07 ^e^	1.13 ± 0.09 ^f^
	Bound	1.34 ± 0.07 ^a^	1.31 ± 0.05 ^a^	1.42 ± 0.04 ^a^	1.67 ± 0.02 ^b^	1.80 ± 0.06 ^bc^	1.96 ± 0.08 ^c^	2.14 ± 0.19 ^d^
	Total	1.70 ± 0.07 ^a^	1.74 ± 0.09 ^a^	1.92 ± 0.09 ^a^	2.28 ± 0.04 ^b^	2.52 ± 0.11 ^b^	2.88 ± 0.14 ^c^	3.27 ± 0.28 ^d^
ABTS	Free	2.22 ± 0.12 ^a^	2.72 ± 0.06 ^b^	2.89 ± 0.12 ^b^	3.34 ± 0.16 ^c^	3.95 ± 0.13 ^d^	4.80 ± 0.06 ^e^	6.14 ± 0.56 ^f^
	Bound	9.33 ± 0.19 ^a^	9.41 ± 0.34 ^a^	9.69 ± 0.32 ^a^	10.43 ± 0.48 ^a^	11.66 ± 0.96 ^b^	13.09 ± 0.67 ^c^	13.98 ± 1.05 ^c^
	Total	11.55 ± 0.22 ^a^	12.13 ± 0.39 ^a^	12.58 ± 0.40 ^ab^	13.77 ± 0.45 ^b^	15.61 ± 0.98 ^c^	17.89 ± 0.69 ^d^	20.12 ± 1.61 ^e^

Data are means ± SD from three independent replicates. Means with different letters in rows indicate statistically significant differences between the different stages of germination (*p* < 0.05).

**Table 2 foods-12-01769-t002:** Contents of the individual bioactive compounds (µg/g DW) in the germinating spelt seeds.

Compound	Content of Benzoxazinoids and Phenolics (µg/g DW) during Germination (h) for the Nongerminated and Germinated Spelt Seeds						
	Nongerminated	12	24	36	48	72	96
Free fraction							
Schaftoside	13.58 ± 1.54 ^b^	9.62 ± 1.22 ^a^	12.3 ± 0.52 ^b^	9.21 ± 0.55 ^a^	12.89 ± 1.40 ^b^	18.53 ± 0.58 ^c^	53.69 ± 2.65 ^d^
Schaftoside isomer 1	46.78 ± 5.41 ^bc^	39.8 ± 5.20 ^ab^	47.94 ± 2.04 ^c^	35.08 ± 1.75 ^a^	46.82 ± 7.44 ^bc^	45.51 ± 1.71 ^bc^	47.48 ± 1.56 ^c^
Schaftoside isomer 2	91.28 ± 15.26 ^b^	83.38 ± 15.77 ^ab^	92.97 ± 3.97 ^b^	71.30 ± 3.00 ^a^	89.71 ± 9.43 ^b^	85.20 ± 5.56 ^ab^	90.42 ± 3.38 ^b^
BOA	ND	ND	4.13 ± 1.32 ^a^	11.88 ± 1.33 ^b^	34.67 ± 3.00 ^d^	19.77 ± 4.41 ^c^	20.49 ± 0.86 ^c^
MBOA	ND	ND	6.98 ± 2.55 ^a^	38.78 ± 5.97 ^b^	111.52 ± 10.38 ^c^	181.05 ± 29.94 ^d^	277.61 ± 15.29 ^e^
HBOA	ND	ND	ND	14.97 ± 6.37 ^a^	26.05 ± 5.72 ^a^	130.08 ± 30.88 ^b^	219.65 ± 46.01 ^c^
*trans*-ferulic acid	1.90 ± 0.19 ^a^	1.90 ± 0.39 ^a^	1.63 ± 0.36 ^a^	2.06 ± 0.46 ^a^	2.05 ± 0.37 ^a^	1.90 ± 0.18 ^a^	3.00 ± 0.70 ^b^
*cis*-ferulic acid	0.96 ± 0.40 ^ab^	1.27 ± 0.33 ^b^	0.95 ± 0.33 ^ab^	1.14 ± 0.47 ^ab^	1.32 ± 0.32 ^b^	0.52 ± 0.13 ^a^	0.52 ± 0.11 ^a^
*trans*-p-Coumaric acid	ND	ND	0.97 ± 0.19 ^a^	1.34 ± 0.14 ^ab^	1.13 ± 0.09 ^a^	1.12 ± 0.14 ^a^	1.59 ± 0.20 ^b^
Bound fraction							
*trans*-ferulic acid	359.92 ± 9.21 ^a^	406.61 ± 12.03 ^a^	419.48 ± 11.44 ^ab^	439.96 ± 32.99 ^ab^	489.16 ± 42.94 ^b^	580.89 ± 38.23 ^c^	753.27 ± 95.87 ^d^
*cis*-ferulic acid	76.69 ± 7.51 ^a^	97.29 ± 14.07 ^a^	93.46 ± 22.90 ^a^	110.44 ± 27.16 ^ab^	140.38 ± 13.47 ^bc^	150.84 ± 17.75 ^c^	203.66 ± 29.30 ^d^
*trans*-p-Coumaric acid	16.87 ± 1.75 ^a^	17.21 ± 0.58 ^a^	18.44 ± 0.81 ^a^	20.18 ± 2.41 ^a^	25.36 ± 1.07 ^a^	50.05 ± 3.69 ^b^	119.77 ± 15.42 ^c^
*cis*-p-Coumaric acid	1.59 ± 0.36 ^a^	1.68 ± 0.23 ^a^	1.73 ± 0.22 ^a^	1.69 ± 0.28 ^a^	2.33 ± 0.29 ^ab^	3.41 ± 0.56 ^b^	9.18 ± 1.56 ^c^

Data are means ± SD from three independent replicates. Means with different letters in rows indicate statistically significant differences between contents in the different stages of germination (*p* ˂ 0.05); not detected (ND).

## Data Availability

The data presented in this study are available on request from the corresponding author.

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
