# Peer review of "Identification and Quantification of Selected Benzoxazinoids and Phenolics in Germinated Spelt (Triticum spelta)"

_foods, 2023, doi:10.3390/foods12091769_

Round 1
Reviewer 1 Report
The paper aims to evaluate the effect of germination on the total phenolic content and antioxidant activity of germinated spelt grains.
The paper presents the results of a comprehensive research and accordingly deserves to be published. The manuscript is a well-structured and results are well analyzed and compared with the results obtained by other researchers. The manuscript includes a balanced introduction, experimental section is clearly described, results and discussion are clearly presented.
The authors stated that the goal of this research was to evaluate the effect of germination on the total phenolic content and antioxidant activity and to provide information that can contribute to the implementation of germinated spelt grains for the production of functional foods. In that sense, suggestion to the authors is to provide some examples of the use of germinated spelt grains (and in what form, whole grain flour, extract, powder, etc.) in food production, especially examples of use in the production of functional food.
Reviewer 2 Report
The abstract is to be revised as it is not showing the issues related to the germination time.
In line 32, Need to write the in vitro and in vivo in italics
In lines 72 to 82, two sentences are contradicting
The introduction has to fine trim, and information on the general germination is given much
Is section 2.1 required? There is the general chemical list and their make. Better to trim it
In the materials and methods check the typographical and editorial issues
Give proper citations for the methods used in the materials and methods section.
Line 246 gives proper notation for +or -
Why given 0 in germination time? it can give as ungerminated? need clarity here soaked but not germinated? Try to clarify in methods
In section 3.1, need to give more citations and should compare with the other previous studies.
What is the C in Figure 2?
Foot Note of Table 1, should write the mean difference row-wise?
Replace " our experiment" with "preset study or in this study"
In images, all the abbreviations should be given in the footnote
The discussion is very good and deep
In conclusion, better to give some value. Also, must mention which germination duration is best.
Fine-tuning the English language is a must
Check for the editorial and typographical errors
The quality of the English is fine, but fine tuning is required.
Reviewer 3 Report
The authors of this manuscript analyzed the germinated spelt for its contents of benzoxazinoids and phenolics.
The manuscript was designed and written in a good way, and it is scientifically sound. It also adds new information related to the bioactive constituents of the spelt grains and how the contents of these compounds were affected by the germination process.
However, there are a few points that need to be corrected.
- Line 113: Triticum spelta should be italic.
- Line 262: delete from.
- Line 301: delete in.
- Line 311: obtained through LC-MS analysis..
- Line 313: 2- Unidentified
- Figure 2, in the free fraction of non-germinated spelt (A): Compounds 13 and 15 are not presented in the figure, while they are quantified in Table 2.
- Figure 2, in the free fraction of 96-hour germinated spelt (B): Compounds 7, 13, and 15 are not presented in the figure, while they are quantified in Table 2.
- Line 321: (BOA, peak 8),
- Lines 321 and 323: Table S2 is not found in the supplementary materials.
- Lines 354-355: (13.58 μg/g DW) is not the lowest concentration of schaftoside in non-germinated spelt, please see the data in Table 2.
- Table 2: there is an error in the statistical letters of Schaftoside isomer 1. How 46.78±5.41 has the letter b, while 45.51±1.71 has the letter bc.
- Table 2, trans-ferulic acid in bound fraction: be sure that 419.48 ± 11.44 has the letter abc.
